# Estimation of Stresses in Concrete by Using Coda Wave Interferometry to Establish an Acoustoelastic Modulus Database

**DOI:** 10.3390/s20144031

**Published:** 2020-07-20

**Authors:** Hanyu Zhan, Hanwan Jiang, Chenxu Zhuang, Jinquan Zhang, Ruinian Jiang

**Affiliations:** 1Klipsch School of Electrical and Computer Engineering, New Mexico State University, Las Cruces, NM 88003, USA; hzhan@nmsu.edu; 2Department of Civil Engineering, University of Wisconsin-Platteville, Platteville, WI 53818, USA; 3Technology and Surveying Engineering, New Mexico State University, Las Cruces, NM 88001, USA; zhuang@nmsu.edu (C.Z.); rjiang@nmsu.edu (R.J.); 4Research Institute of Highway, Ministry of Transport, Beijing 100088, China; jq.zhang@rioh.cn

**Keywords:** coda wave interferometry, acoustoelastic modulus, Kaiser effect, stress recovery in concrete, nondestructive testing

## Abstract

This article presents an experimental study of estimating stresses in concrete by applications of coda wave interferometry to establish an acoustoelastic modulus database. Under well-controlled laboratory conditions, uniaxial load cycles were performed on three groups of 15 × 15 × 35-cm concrete prisms, with ultrasonic signals being collected continuously. Then, the coda wave interferometry technique, together with acoustoelastic and Kaiser theories, are utilized to analyze the stress-velocity relations for the distinct ranges before and after historical maximum loads, forming an acoustoelastic modulus database. When applied to different concrete samples, their stresses are estimated with a high degree of accuracy. This study could be used to promote the development of novel nondestructive techniques that aid in structural stress monitoring.

## 1. Introduction

The importance of concrete to the global infrastructure is evident by its pervasiveness in the world. Mechanical load is one of the most frequently evoked behaviors that degrade the structural integrity. For the sake of both economy and safety, evaluating stress states in concrete structures such as bridges is in great demand.

Concrete is a multi-component conglomerate comprised of cement, sand grains, stones and porosities. The property of each composite under variable environmental effects, together with geometrical properties, construction tolerances and damages, contributing different difficulties in the stress analysis. Current techniques usually rely on surface-placed and/or pre-implanted gauge arrays to collect strain profiles at their installation positions and then estimate strain diagrams through intercept algorithms. These techniques, in general, are time-consuming, cumbersome and costly. For example, pre-implanted methods are not available for a large number of existing structures being in service over many years. Gauges may suffer irreversible deformations when significant cracking or medium changes take place nearby. Furthermore, gauges are only capable of detecting stress variations at locations where they are exactly placed, and therefore, any nonlinear variations or damages appearing among them will significantly bias estimation results, even if plenty of gauges are installed closely [1,2].

Ultrasonic waves constitute one of the primary tools to monitor concrete elements, mostly on laboratory scales. Measuring the velocity, amplitude and nonlinearity of direct ultrasound waves shows promise for detecting the thickness, delamination and damages [3,4]. However, a high heterogeneity in concrete causes strong multiple scatterings during wave propagations. As a consequence, the wave energy is significantly attenuated in a process similar to heat diffusion. The energy is not lost but mostly transformed into long-lasting waveforms arriving later in the record, forming the so-called coda waves [5]. These features are well-known to severely affect the performance of direct wave-based detection techniques. Fortunately, it has been demonstrated that coda wave is very sensitive to changes occurring in the large volume of a medium [5,6,7,8,9,10,11,12,13]. This sensitivity is attributed to the fact that coda waves propagate following complex trajectories not limited to the direct path between sensors, and they may traverse the regions of interest repeatedly. Thanks to this property, coda waves have been successfully used for several nondestructive testing applications, such as structural health monitoring [5,6,7] and crack characterization [8,9,10,11,12,13].

It should be noted that the techniques mentioned above [5,6,7,8,9,10,11,12,13] are designed to detect and characterize damages. The evaluation of the stress status in concrete remains a great challenge. In literature, many experiments have shown that coda waveforms are varied with external stresses, but few works provide qualitative and quantitative insights into their correlations [14]. Recently, coda wave interferometry (CWI), a technique initially developed from seismology [15], was introduced to civil engineering [16]. It is based on the comparison of the perturbed and reference coda waves to precisely evaluate the propagation velocity variations *dv/v* up to the order of 0.001% [17]. Following the CWI technique, Larose et al. observed the almost linear stress-*dv/v* dependences in the measurements. Then, the acoustoelastic theory was used as the physical basis to describe this relation [17]. Later on, Shokouhi et al. and Stahler et al. found that the slopes of the dependence, now named the acoustoelastic coefficients, are not fixed when cyclic loads were conducted on a concrete sample [18,19]. Zhang et al. modified the acoustoelastic theory by adding the Kaiser effect to explain the slope changes caused by stress-induced microcracks [20]. Even though these results are very encouraging, their works focus on the theory development to explain stress-velocity dependences. Further studies are still necessary for extending coda waves to stress estimations. For example, their actual slope values depend on time window selections in coda signals, which complicate data processing and lose the generality of CWI applications on stress estimations. In addition, these experimental tests attempt to fit external loads and coda data through appropriate slopes, not recover stresses in concrete. 

In this paper, we present an experimental study of establishing an acoustoelastic modulus database for stress estimations in concrete. According to the design specifications, three groups of concrete samples were casted with different strength grades. Under well-controlled laboratory conditions, identically uniaxial load cycles were performed on each sample, where ultrasonic signals were collected continuously with source-receiver pairs. A stepwise CWI technique was applied to the full coda waveforms during the propagation period (i.e., no time-window selection) to quantify their velocity variations caused by external loads and stress-induced micro-damages. Following the acoustoelastic theory, for each group, a concrete specimen was used to calculate its best-fitting slopes of the stress-velocity relations. Due to the Kaiser effect, two slopes were individually obtained for the distinct ranges before and after the historical maximum loads, forming an acoustoelastic modulus database. Then, the moduli were applied to the remaining concrete samples, and their stresses were recovered with a high degree of accuracy. The applicability of a coda wave on the stress estimation was demonstrated for different types of concrete and different samples comprised by the same concrete. In addition, the modulus database showed several trends, e.g., the modulus values descended as the concrete strength grades increased and severe damages broke the stress-velocity dependence. The potential applications of these findings are also discussed for extending the approach to real infrastructures. So far, there is no mature nondestructive technique of measuring stresses in concrete structures, and therefore this study could be very valuable for nondestructive testing and structural health monitoring.

## 2. Experimental Setup

### 2.1. Concrete Samples

According to the design specification JGJ55-2011, three types of ordinary concrete, corresponding to the strength grades of C30, C40 and C50, were made with cement, water, sand grain, stone and superplasticizer. Their mix proportions and mechanical properties are listed in Table 1. For each type of concrete, 4 identical specimens with dimensions of 15 × 15 × 30-cm were casted and maintained in a curing room of 20 ± 2 °C and 95% humidity (GB/T50081-2002 standard) for several weeks. After maintenance, one specimen from each type was selected randomly to examine its ultimate strength through a creep test. The measured results agreed with their theoretical break-down stresses, indicating that the productive processes well follow the design specifications. Then, as shown in Figure 1a, the remaining specimens were numbered in groups prepared for the later experiments, e.g., the label C30-2 denotes the second sample of the concrete type C30. 

### 2.2. Measurement Procedure

All the concrete specimens experienced the identical procedures in a climate-controlled room at the National Engineering Laboratory for Bridge Structure Safety at Beijing, China. That is, their experimental environment (temperature and humidity), sensor arrangement, loading step and measurement setup were exactly the same. 

Ultrasonic transducer RS-2A with diameter 18.8 mm and thickness 15 mm had a broadband working frequency of 60~400 kHz. For each sample, 4 transducers, worked as a source (S1) and three receivers (R1-R3), were glued on its opposite sides using the coupling medium of high vacuum silicone grease. Figure 1b,c showed their location details. S1 and R2 were placed in the center of the left and right surfaces to collect data mainly corresponding to the horizontal effects, whereas the S1-R1 and S1-R3 pair observations contained both horizontal and vertical effects. In addition, 6 strain gauges were also installed around the transducers. The gauge measurements were accomplished with a DH3821 multichannel static strain test and analysis system, and their data were compared to the applied and estimated stress values in Section 4.

Each concrete sample was loaded in uniformly uniaxial presses by a YEW-5000F1 hydraulic pressure testing machine that contains a ball seat for leveling. Thus, the values of the applied forces and stresses hold a straightforward relationship, i.e., stress = force/contact area 15 × 15 cm^2^. Three loading-unloading cycles with a total of 100 steps were conducted, see Figure 2. For convenience of description, they are indicated by the characters encountered in alphabetical order. The applied forces firstly increased to 50 kN and then decreased back to 0 kN with a fixed step of 5 kN in cycle 1 (A–C). The second cycle (C–E) repeated this loading-unloading steps, but the interval was set as 10 kN. Finally, large stresses up to 800 kN were applied and released in cycle 3 (E–G).

Each loading step lasted 2 min, where the first 1 min was utilized to stabilize the stress conditions in concrete before data collections. During the 2nd minute, a high-power ultrasonic system RITEC-SNAP-5000 was connected to the source to emit the identical excitation impulse at 400 kHz every 0.1 s. Note that a higher frequency leads a great sensitivity to weak changes but, meanwhile, enhances the energy attenuations due to scatterings and dissipations. This frequency choice is a trade-off result between the detection capability and sensitivity, and it is consistent with the typical frequencies used to generate coda waves in concrete [14]. In addition, using single-frequency impulses, as opposed to transient bursts, can simplify the data processing and increase the transmission range. Each receiver continuously collects the response signals at a 5-MHz sampling frequency that conforms to the Nyquist theorem. The signals were firstly amplified by a Smart AE amplifier and then analyzed with a multichannel MSO4104B-L oscilloscope and a high-performance desktop. Each source-receiver pair had an independent time trigger and counter to accurately record their excitation and observation times. A photograph and schematic layout of the measurement procedures are shown in Figure 3. 

In a word, each step included three sets of data collected with different source-receiver pairs, and each set contained 600 (i.e., 1 min/0.1 s) full coda signals. To improve the signal-to-noise ratio (SNR), for each set the 600 acquisitions were averaged to obtain the mean waveform. These mean waveforms were recorded in the form of $\;E_{s,r\;}^{l}$, where *s*, *r* and *l* were the relevant numbers of the concrete sample, receivers and loads. For example, $\;E_{c30-1,1\;}^{1}\;$ denotes the average waveform collected from the C30-1 sample by the R1 receiver at the 1st loading step (5 kN).

## 3. Methodology and Data Processing

### 3.1. Velocity Variations Determination

A coda wave observed at a given position is a sum of the partial waves from all possible trajectories, and its energy is transported in a process similar to heat diffusion. The coda wave travels a large volume for a long time so that small changes anywhere in the concrete result in a notable waveform variation. Figure 4 presents an example of two coda waveforms, $E_{c30-1,2\;}^{0}\;$ and $E_{c30-1,2\;,}^{4}\;$ measured with the same sample and receivers at different loads. Their signals in three time windows are enlarged to indicate the variation details. Windows 1 and 2-3 correspond to the single-scattering (i.e., direct wave) and diffusion (i.e., coda wave) regimes, respectively. As the loads increase, the coda waveforms show the compression effects. With sufficient multi-scattering behaviors, this effect becomes more obvious in the late part of the signals (window 3), whereas the direct waves are nearly invariable. These time compressions or dilations in the coda waves can be considered as the relative velocity variations *dv*/*v* and quantified by code wave interferometry (CWI) techniques. The CWI expression is generally given by [16]
(1)$$CC=\frac{\int_{t_{1}}^{t_{2}}E_{2}\left[t\left(1+\varepsilon \right)\right]E_{1}\left(t\right)dt}{\sqrt{\int_{t_{1}}^{t_{2}}{\left\{E_{2}\left[t\left(1+\varepsilon \right)\right]\right\}}^{2}dt\int_{t_{1}}^{t_{2}}{\left[E_{1}\left(t\right)\right]}^{2}dt}}\;;\;\varepsilon =\frac{dv}{v}\;$$

The cross-correlation *CC* indicates the degree of similarity of two signals *E*_1_ and *E*_2,_ in a chosen time window [*t*_1_, *t*_2_]. The dilation rate *ε* computes an increase or decrease of the propagation velocity, and its value is determined by maximizing the *CC* values. Interpreting the *ε* and *CC* relations can distinguish different types of changes [16]. If the *CC* remains a large value by changing the time lag, this means that the wave velocity is modified homogeneously in a large region due to stress changes. On the other hand, if the *CC* keeps at low values no matter what *ε* value is chosen, this is because of an occurrence or aggravation of cracks that change the medium impedance contrast. In this study, we focused on the first type of stress changes, but *CC* values in the second case can be worked as an indicator of internal damage levels. 

As shown in Figure 4, the compression extents in windows 2 and 3 have a little difference, implying *dv/v* values of coda waves depend on time [19,20]. For crack characterization, selecting appropriate time windows through trials, although cumbersome, is capable of improving the detection accuracy, such as the locations and depths of microcracks [8,9,10,11,12]. However, in real-life situations, the factors of composite, mixture, geometry, size, construction and corrosion contribute different uncertainties in concrete properties. Thus, it is impossible to set a universal time window suitable for all kinds of concretes and structures. As a result, *dv/v*-stress dependences are time-varying, which prevents coda waves (*dv/v*) as references for stress estimation applications.

Here, a simple way, by applying stepwise CWI [21] to a full waveform, is used to overcome this issue. In particular, we directly calculated *dv/v* values using full coda waveforms instead of a specific part (no time window) to remove the time dependence [7]. However, this operation will slightly reduce the sensitivity, since *dv/v* now is the average of the velocity variations during the whole propagation period *T* that includes the earlier parts of signals with smaller phase changes. To further solve this problem, stepwise velocity variations are calculated by using just the previous measurement (i.e., same sample and same receiver but previous load) as the reference and then multiplied to provide information on the cumulative changes relative to the beginning (step 0) of the experiments [21]. With these modifications, CWI now is expressed as
(2a)$$CC_{s,r}^{l}=\frac{\int_{0}^{T}E_{s,r}^{l}\left[t\left(1+\varepsilon_{s,r}^{l}\right)\right]E_{s,\;r}^{l-1}\left(t\right)dt}{\sqrt{\int_{0}^{T}{\left\{E_{s,r}^{l}\left[t\left(1+\varepsilon_{s,r}^{l}\right)\right]\right\}}^{2}dt\int_{0}^{T}{\left[E_{s,r}^{l-1}\left(t\right)\right]}^{2}dt}}\;;\;\varepsilon_{s,r}^{l}=\frac{dv_{s,r}^{l}}{v_{s,r}^{l-1}}\;$$
and
(2b)$$\frac{dv_{s,r}^{l}}{v_{s,r}^{0}}≅\left(1+\varepsilon_{s,r}^{l}\right)\left(1+\varepsilon_{s,r}^{l-1}\right)\cdots \left(1+\varepsilon_{s,r}^{1}\right)-1$$

### 3.2. Acoustoelastic Modulus Calculation

Concrete is a multicomponent material exhibiting nonlinearly elastic property, and its stress-strain (*σ*-*ε*) relation contains quadratic terms to modify the classic Hook’s law [22]:(3a)$$\sigma_{ab}=C_{abcd}\varepsilon_{cd}+C_{abcdef}\varepsilon_{cd}\varepsilon_{ef}$$

*C_abcd_* is the modulus of Hook’s law that can be expressed by the two Lame parameters, λ and *u*. The tensor *C_abcdef_* has three independent Murnaghan constants (*l*, *m* and *n*) to describe the second-order effect of concrete in the Lagrangian coordinates. Based on Equation (3a), the stress-velocity dependence in an acoustoelastic effect has been derived [23]: (3b)$$v_{ij}^{k}={\left(v_{ij}^{k}\right)}^{0}\left(1+\beta_{ij}^{k}\sigma_{k}\right)$$

Here, *i*, *j* and *k* represents the directions of the wave propagation, polarization and stress field, respectively. The acoustoelastic coefficient *β* depends on the tensors of *C_abcd_* and *C_abcdef_*. *v*^0^ is the velocity in the initial state (i.e., intact and stress-free) of concrete. Equation (3b) is initially developed for a direct wave. However, for coda waves, multiplying scattering behaviors cause strong conversions between different polarized and directional wave types. Consequently, CWI provides the analysis of the average velocity rather than the velocity of each specific type of wave [19,20,24], especially for this study, where full coda waveforms and small concrete specimens are used. In addition, the uniaxial loads along a fixed direction are performed during our experiment; thus, the acoustoelastic coefficient takes the form of
(4)$$\beta_{s,r}^{l}=\frac{v_{s,r}^{l}-v_{s,r}^{0}}{v_{s,r}^{0}}=\frac{dv_{s,r}^{l}}{v_{s,r}^{0}}/\sigma^{l}$$

Figure 5 shows an example of the *dv/v* calculation values as a function of the applied stresses, corresponding to the C30-1 and R1 acquisitions. During stressing from the initial state A to the quasi-brittle state Q, the observations of the approximately linear stress-*dv/v* dependences are consistent with the acoustoelastic model in Equation (4). The slopes *β* are varied in some trends: (1) the velocity change *dv/v* at completely releasing states C and E is lower than the initial state A, i.e., *dv/v* values at state E < C < A. (2) The *dv/v* curves of the later loading cycles rise faster than the previous loading cycles, and they intersect at the historical maximum load. For example, the curves of the 2nd loading cycle (C-B), with the steeper slope, cross the 1st loading cycle (A-B) at its maximum load (50 kN). Similarly, the E-D curves cross the 2nd loading cycle C-D at a 100-kN load. (3) After the historical maximum loads, the curve climbs up from point B to D following the similar slope of the A-B curve, and the diagram from point E to D develops along the path that shows a similar slope to the C-B curve.

The slope variations are consistent with the Kaiser effect that describes “stress memory” behaviors in concrete [20,25]. Specifically, concrete has the capability of remembering the previous maximum stress level ever experienced. From a microscopic view, this is because of new microcracking that takes place only if the load exceeds its historical maximum level; otherwise, no significant modification in the microstructure will occur. At states A–B and B–D, where the load level exceeds the previous maximum, the acoustoelastic behavior increases the propagation velocity, but the microcracking causes a decrease in velocity. Consequently, their combination effects raise the propagation velocity slowly. However, at states C–B and E–D (do not exceed), no new microcracks are created, and the faster velocity increase is mainly due to the acoustoelastic effect. 

After state Q, the velocity continues to fall even for the increasing stresses. Concrete enters in the quasi-brittle regime where microcracks develop rapidly in length, width and numbers, and begins to connect to form irreversible larger fissures. Reaffirm that this study focuses on the stress estimation, and therefore *dv/v*-stress relations are only applied before a quasi-brittle state. That is, the upper bounds of the stress estimation levels for the concrete samples used here are 20~27 MPa, depending on their strength grades. However, the data corresponding to later loading steps show great sensitivity to the microcracking process, which provides a potential tool to detect defects for future works. 

## 4. Stress Estimation and Results

Based on Equations (2) and (4), for each type of concrete, a specimen (C30-1, C40-1 and C50-1) is firstly used to calculate the best-fitting slopes (acoustoelastic modulus $\;\beta_{s,r}^{l}$) through the standard least square algorithm. Due to the Kaiser effect, two slopes are individually calculated for the distinct ranges before and after the historical maximum loads (BHML and AHML), forming a modulus database, as shown in Table 2. Then, the stresses in other samples (C30-2/3, C40-2/3 and C50-2/3) are estimated by applying the moduli to their coda wave measurements. We note that Table 2 lists the modulus values corresponding to all receivers and states for evaluation and analysis, but the database need not be so complicated. For each type of concrete, only moduli corresponding to the AHML and BHML states from a source-receiver pair are necessary.

The AHML and BHML moduli in Table 2 are calculated with all states where the historical maximum load has and has not been exceeded, respectively. Their values agree well with the moduli of the associated substates, i.e., the moduli of A-B, B-D and AHML indicate very similar values, as well as the moduli of C-B, E-D, and BHML indicate very similar values. This phenomenon confirms the consistency of the acoustoelastic effects in a broad load range [19], indicating only the AHML and BHML moduli are required to establish databases for the stress estimation. In addition, the modulus of the concrete descends in value in the following order: C30, C40 and C50, whereas the stress values of the quasi-brittle state Q rise with the same order. These trends are in agreement with our expectations that concrete with a high strength grade owns a low elasticity and large carrying capability. 

Another finding is that the modulus values for the same concrete type but different receivers show very little difference. It seems that the R2 moduli tend to be slightly smaller than the R1 and R3 moduli, but this trend is not obvious. As shown in Figure 1, S1 and R2 are placed at identical heights to collect data corresponding to horizontal effect, and the S1-R1 and S1-R3 pair acquisitions contain the effects from both horizontal and vertical directions. In theory, waves polarized parallel to the stress vector show greater sensitivity to the stress variation than those with a perpendicular polarization. However, multiple scattering causes strong energy conversions between differently polarized and directional wave types. Especially for full coda waveforms and small-size specimens used in this work, the sufficient scatterings inside the volume and at boundaries further intensify this conversion. As a superposition of multiplied scattered waves from all trajectories, the coda wave contains energy contributions from all types of waves. Furthermore, the distance difference between the S1-R1/S1-R3 and S1/R2 pairs is small compared to their horizontal spacing. As a result, the moduli represent average results of all the wave types and directions according to the prevalence in the coda wave field. This property may benefit the application of this approach on real structures, since it provides some flexibility and fault tolerance for sensor positions.

Then the stress values in other concrete specimens were estimated by using the modulus database. The stress estimation results for the C30-2, C40-2 and C50-2 samples, corresponding to the R2 collections, are shown in Figure 6 as an example. The applied forces and the stress values measured by strain gauges are displayed for comparison. The coda wave-based estimation results agree well with the real and measured values. For other receivers (R1 and R3) and samples (C30-3, C40-3 and C50-3), the stress values are also recovered accurately, and they are not indicated here for the sake of brevity.

## 5. Conclusions and Discussion

In conclusion, we present and demonstrate the application of a coda wave to estimate stresses in concretes. By using coda wave interferometry and the Kaiser theory, a database is established by incorporating the acoustoelastic modulus of different concrete types and loading ranges. When applied to concrete samples, their stress results are estimated with a high degree of accuracy. This study could be very valuable for nondestructive testing and structural health monitoring.

It should be noted that this work is conducted in an indoor laboratory environment. There are several issues worthy of discussion for extending the approach to real concrete structures. Table 1 lists the modulus corresponding to a single type of pure concrete. However, a real structure may be comprised by multiple kinds of concretes and nonconcrete components, such as steel bars, as well as the casting and construction processes may not be enacted as strictly according to the design specifications as our operations in the laboratory. In addition, temperatures throughout the day and year impose changed thermal effects. All of these factors will contribute variations in the modulus values. Perhaps the most straightforward solution is to measure the modulus of each component (e.g., each type of concrete and steel) as a function of temperature and then determine a reasonable range of modulus values based on the structural mixture and temperature conditions. This process is similar to the classical procedure of making a table of physical parameter such as refractive index and elasticity. A simpler alternative is to only measure the moduli of key components and key temperatures and estimate the mixture modulus through interception and machine-learning algorithms.

As noted in Section 3 and Section 4, two individually acoustoelastic moduli are required for the distinct ranges before and after the historical maximum loads. For a large number of existing concrete structures with long-term service, perhaps the AHML modulus is more useful since the daily loads normally are smaller than their historical maximum values. The maximum loads can be estimated from annual check reports, historical records, safety/serviceability requirements and cracking states.

Stresses conducted on large structures sometimes can be highly nonuniform, anisotropy and dynamic, e.g., traffic through a bridge. Then, the velocity changes induced by stresses can also be highly anisotropy. Even though coda wave is a mixture of differently polarized and directional wave types, its velocity variations may still exhibit a directional prevalence and preference. Consequently, modulus values may slightly vary depending on source-receiver locations. In this case, determination of the moduli needs to measure the component values in coordinates and then combine them in three-dimensional regions based on stress and wave relative directions. Fortunately, real structures usually have certain parts that mainly respond to stresses in one direction. In this case, the modulus calculation and stress estimation can be simplified by placing source-receiver pairs appropriately to eliminate position dependence. Furthermore, between the elastic and quasi-brittle states, there is an intermediate regime where the wave velocity slowly increases without any change in the applied force [26]. This slow dynamic effect will not affect our modulus and stress results in the elastic regime, but it will cause a time-dependent modulus in the slow dynamic regime. Understanding of this effect may extend this approach’s availability for broader stress ranges. In literature, only very few works involve anisotropic and slow dynamic effects on coda waves, and future studies on these two issues are required.

## Figures and Tables

**Figure 1 sensors-20-04031-f001:**
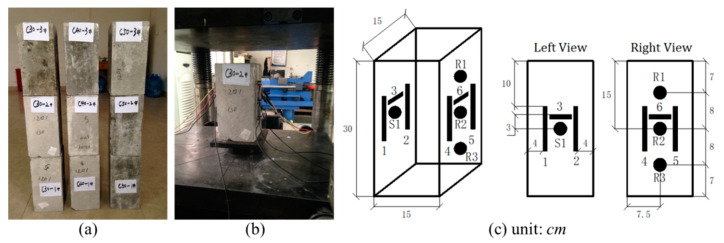
(**a**) Concrete samples, and (**b**,**c**) photographs and schematic layouts of the transducer (black solid circles) and gauge (black solid rectangles) locations.

**Figure 2 sensors-20-04031-f002:**
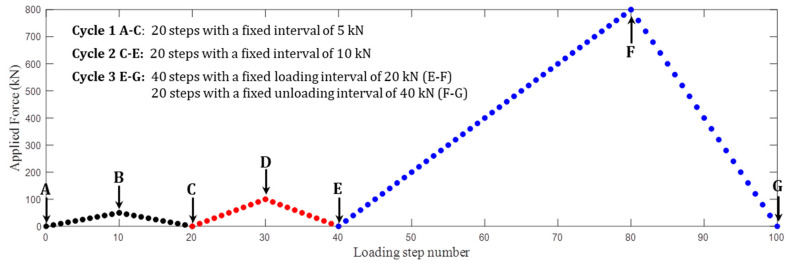
The loading-unloading steps conducted on each concrete sample.

**Figure 3 sensors-20-04031-f003:**
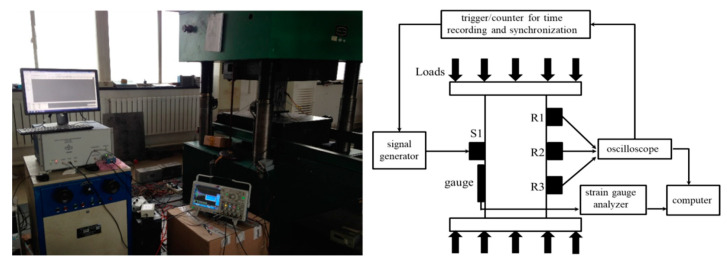
A photograph and schematic layout of measurement procedure.

**Figure 4 sensors-20-04031-f004:**
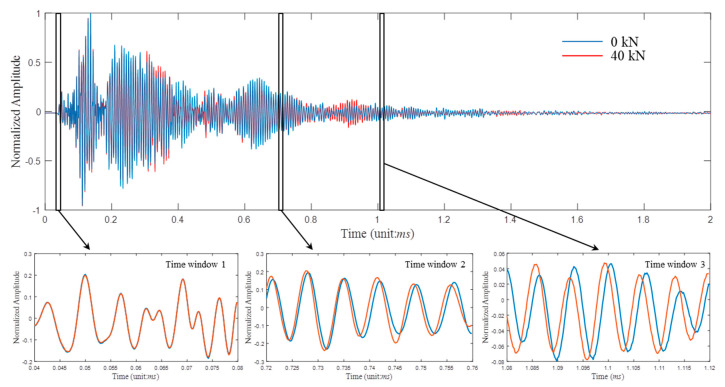
The average full coda waveforms collected from the C30-1 sample by the R1 receiver at 0 and 40 kN.

**Figure 5 sensors-20-04031-f005:**
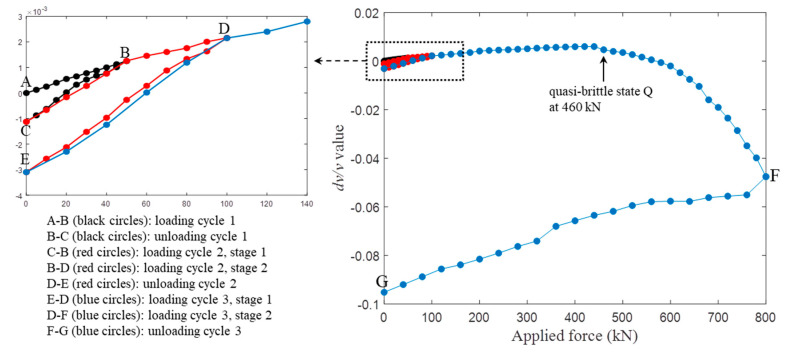
An example of the *dv/v* calculation values as a function of the applied forces, corresponding to the C30-1 and R1 acquisitions. The characters A–G denote the associated loading steps (see Figure 2).

**Figure 6 sensors-20-04031-f006:**
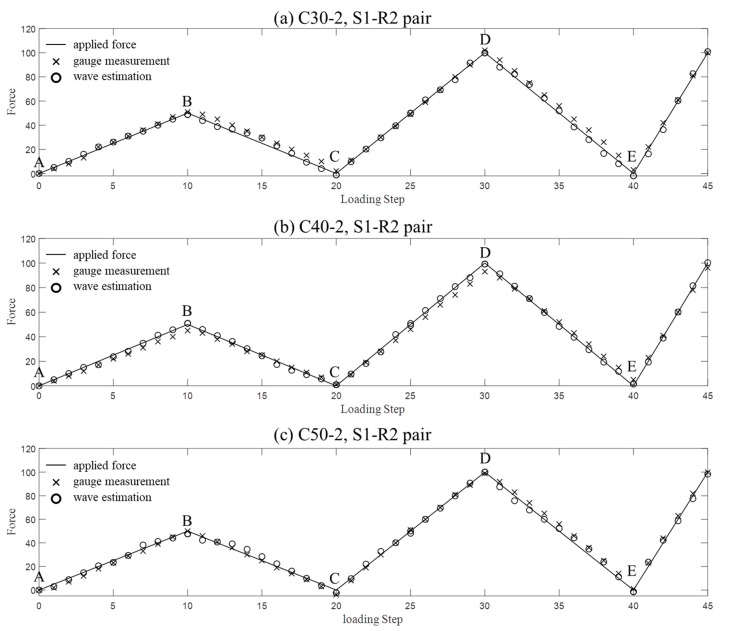
The applied (line), measured (x) and estimated (circle) stress values for the concrete samples of: (**a**) C30-2, (**b**) C40-2 and (**c**) C50-2.

**Table 1 sensors-20-04031-t001:** The mix proportion and mechanical properties.

	Cement (kg/m^3^)	Water (kg/m^3^)	Sand (kg/m^3^)	Gravel (kg/m^3^)	Superplasticizer (kg/m^3^)	Break-Down Force (kN)
C30	360	148	789	1090	3.6	1000
C40	400	139	706	1154	4.3	1200
C50	400	131	700	1136	4.8	1300

**Table 2 sensors-20-04031-t002:** A database of the concrete acoustoelastic modulus *β* (unit: 10^−4^/MPa).

Type	Receiver	A–B	B–D	AHML (A–B–D)	C–B	E–D	BHML (CB, ED)	Q (kN)
C30	R1	5.623	6.073	5.852	10.685	11.365	11.030	460
	R2	5.175	5.848	5.522	10.350	10.348	10.350	460
	R3	5.573	5.852	5.681	11.700	11.246	11.475	460
C40	R1	4.563	4.619	4.592	9.520	8.917	9.223	540
	R2	4.502	4.280	4.392	8.780	9.446	9.119	540
	R3	4.725	4.277	4.507	9.225	9.560	9.394	540
C50	R1	3.823	3.821	3.823	7.313	7.142	7.227	600
	R2	3.600	3.602	3.601	6.975	7.533	7.259	600
	R3	3.602	3.827	3.717	7.425	6.975	7.205	600

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
