# Peer review of "Estimation of Stresses in Concrete by Using Coda Wave Interferometry to Establish an Acoustoelastic Modulus Database"

_sensors, 2020, doi:10.3390/s20144031_

Round 1

Reviewer 1 Report

The innovation of this study is using the whole period to estimate the cross-correlation factor in  applying CWI instead of using certain time period. Furthermore, the equation to estimate acoutoelastic coefficient is also newly defined. The method is logical and the experimental results are promising . However, this method seems only applicable at the low stress level. Can authors clarify the upper bound of the stress level which can reasonably estimate the stress level using the present coda wave method with respect to the ultimate failure stress.

Equation 4 indicates the relationship to obtain the acoustoelastic coefficient is established between dv/v and stress. But in either Figures 5,6 or Table 2, the acostoelastic modulus β is in the unit of 1/Force, not 1/stress. As a result, the modulus is not applicable to the other specimen with different geometry or loading type even for the same mixture of concrete experiencing the applicable stress level. The authors should explain the reasons to demonstrate the experimental results in such way.

Author Response

We would like to thank the reviewer for the comments and suggestions. Their critiques certainly helped to make this paper stronger.

Based on the comments from both reviewers, we made several changes that are highlighted with red color in the new work. Table 2 was reworked by replacing the results with a unit of MPa. The detailed responses corresponding to each comment are shown in the attachment.

Reviewer 2 Report

A very interesting topic and article,

there are following points that are not clear for me or I missed .

1.) do the aleatory uncertainties effect the measurments significant and how do you treat with respect to the reproduceable of measurements the statistiscal uncertainties?

2.) how do you compensate temperature effects, randomness of stiffness and homogenity in concrete surfaces

Author Response

(The authors gave the same response as above.)
